# Correlates of Children’s Independent Mobility in Canada: A Multi-Site Study

**DOI:** 10.3390/ijerph16162862

**Published:** 2019-08-10

**Authors:** Negin A. Riazi, Sébastien Blanchette, François Trudeau, Richard Larouche, Mark S. Tremblay, Guy Faulkner

**Affiliations:** 1School of Kinesiology, Faculty of Education, The University of British Columbia, Vancouver, BC V6T 1Z4, Canada; 2Département des sciences de l’activité physique, Université du Québec à Trois-Rivières, Trois-Rivières, QC G8Z 4M3, Canada; 3Faculty of Health Sciences, University of Lethbridge, Lethbridge, AB T1K 3M4, Canada; 4Healthy Active Living and Obesity Research Group, Children’s Hospital of Eastern Ontario Research Institute, Ottawa, ON K1H 8L1, Canada; 5Centre for Hip Health and Mobility, Vancouver Coastal Health Research Centre, Vancouver, BC V5Z 1M9, Canada

**Keywords:** active transportation, built environment, socioeconomic status, physical activity, social–ecological framework, urbanization

## Abstract

Globally, physical inactivity is a concern, and children’s independent mobility (CIM) may be an important target behavior for addressing the physical inactivity crisis. The aim of this study was to examine correlates of CIM (8–12 years old) in the Canadian context to inform future interventions. CIM was measured via parent surveys. Individual, social, and environmental correlates of CIM were examined using a social–ecological framework. 1699 participants’ data were analyzed using descriptive statistics and gender-stratified linear mixed-effects models while controlling for site, area-level socioeconomic status, and type of urbanization. Individual correlates including child grade (*β* = 0.612, *p* < 0.001), language spoken at home (*β* = −0.503, *p* < 0.001), car ownership (*β* = −0.374, *p* < 0.05), and phone ownership (*β* = 0.593, *p* < 0.001) were associated with CIM. For boys, parental gender (*β* = −0.387, *p* < 0.01) was negatively associated with CIM. Parents’ perceptions of safety and environment were significantly associated with CIM. Location (i.e., site) was significantly associated with CIM (ref: Trois-Rivières; Ottawa (*β* = −1.188, *p* < 0.001); Vancouver (*β* = −1.216, *p* < 0.001)). Suburban environments were negatively associated with boys’ independent mobility (*β* = −0.536, *p* < 0.05), while walkability (400 m *β* = 0.064, *p* < 0.05; 1600 m *β* = −0.059, *p* < 0.05) was significantly associated with girls’ independent mobility only. Future research and interventions should consider targeting “modifiable factors” like children’s and parents’ perceptions of neighborhood safety and environment.

## 1. Introduction

The World Health Organization has deemed physical inactivity a “global public health problem” [1] as evidence suggests that most children in many countries are insufficiently active [2]. Achieving adequate levels of physical activity not only benefits cardiovascular and bone health [3] but also improves brain function and mental health (e.g., anxiety, depression) [4,5,6,7]. The level of moderate- to vigorous-intensity physical activity (MVPA) among Canadian children has not changed significantly over the course of nearly a decade (2007 to 2015) [1]. According to the Canadian Health Measures Survey only 7% of Canadian children and youth are accumulating 60 min of MVPA on at least 6 out of 7 days, with 33% achieving a weekly average of at least 60 min MVPA daily [8]. If physical activity levels have failed to increase over the past decade despite initiatives aimed at trying to encourage higher levels of physical activity (e.g., interventions have been largely school-based, and in general, interventions may not have been implemented at a sufficient scale) [9], it is important to consider alternative approaches in helping children achieve the recommended levels of physical activity. There remains a limited understanding regarding the determinants of change in children’s physical activity [10]; children’s independent mobility (CIM) could be one such determinant. Additionally, CIM may provide other developmental benefits for school-aged children.

CIM is defined as a child’s freedom to travel and play around their neighborhood or city without parental supervision [11,12]. Previous research has shown that children with higher levels of independent mobility tend to be more active [13,14,15,16]. Beside the benefit of increased physical activity, CIM can also provide a number of personal, psychosocial, and developmental benefits. CIM can help improve risk assessment, spatial awareness, and wayfinding skills and help in “processing and structuring environmental knowledge” [12,17], p. 65. Additionally, greater CIM provides an opportunity to interact with peers, improve self-confidence, develop better decision-making skills, and gain the competence to navigate their neighborhood safely [12,16,18,19].

Of concern is the dramatic decline in CIM levels worldwide over the last few decades [20,21,22]. A report by Shaw and colleagues [20] reported that CIM levels varied worldwide and significant restrictions (e.g., what children are allowed to do, how far children can roam) were placed on CIM in nearly all 16 countries surveyed. While some countries like Finland, Germany, and Norway had higher aggregate rank scores of CIM compared to other countries (e.g., France, Sri Lanka, Brazil, Ireland, and Australia), overall the report found low levels of CIM internationally. Although various studies have examined correlates of CIM internationally, few studies have examined CIM in Canada. In general, Canadian studies examining CIM have had a narrow geographic scope [23,24,25]. CIM and the correlates of CIM likely vary across locations (e.g., cities), urbanization (e.g., urban, suburban, rural), and socioeconomic status (SES; e.g., low vs. high). Therefore, sampling should reflect that potential variation to ensure that the role of perceived and objectively measured built environment influences in particular is not underestimated.

Research on correlates of any type of physical activity, including CIM, can be mapped through a social–ecological framework [26,27]. The framework emphasizes the dynamic interplay between multiple spheres of influence on a health behavior. These include individual correlates (e.g., child’s age, child’s gender, SES), social environment correlates (e.g., social norms, perceptions of neighborhood), built and physical environment correlates (e.g., urbanization, walkability), and policy correlates. Guided by the social ecological framework, the aim of this study was to examine the individual, social, and physical or built environmental correlates of CIM in grade 4, 5, and 6 children (10–12 years old) in Canada.

## 2. Materials and Methods

### 2.1. ATIM Study

This cross-sectional study is based on data drawn from the ATIM project (Active Transportation and Independent Mobility). The ATIM project was a large, national, school-based study conducted across three regions in Canada (Ottawa, Ontario; Vancouver, British Columbia; and Trois-Rivières, Québec) with a primary aim of examining active transportation, CIM, and physical activity in elementary-aged school children. Canadian census data show wide variability in population size, climate, language, and ethnic make-up across these three regions [28]. The secondary aims were to examine the correlates of active transportation, independent mobility, and physical activity. Ethics approval for conducting the ATIM project was obtained from the Research Ethics Boards at Children’s Hospital of Eastern Ontario (15/103X), University of British Columbia (H15-02710), and Université du Québec à Trois-Rivières (CER-15-218-07.05), as well as from participating school boards.

### 2.2. Participants

A purposive sampling strategy was used to recruit schools in each site. A priori power calculations identified that a sample of at least 1080 children (8–12 years old) was needed to examine correlates of CIM. Overall, 1892 child–parent dyads (Ottawa: 510; Vancouver: 828; Trois-Rivières: 554) in grades 4, 5, and 6 were recruited from elementary schools in Ottawa (*n* = 12), Vancouver (*n* = 13), and Trois-Rivières (*n* = 12). The schools were stratified according to urbanization level (urban, suburban, rural) and socioeconomic level (low, high). Once school board approval was obtained, grade 4, 5, and 6 classes were recruited. As some classes were mixed grade, a few grade 7 students also participated in the study. For participation in the study, written consent was obtained from school officials and parents and assent from the children. Parents and children were asked to complete a survey on CIM and active transportation, which was available in both English and French languages, and return to the school one week later. Additionally, the Vancouver site translated the parent and child surveys to Punjabi and Mandarin in an effort to be more inclusive of the diversity in the Greater Vancouver Area. Back translations were conducted to ensure the accuracy of the translations. Data collection took place between March 2016 and June 2017. Overall, the study had a 54% consent rate and 97% participation rate across the three sites. The final sample used for the analysis included data from 1699 participants who returned either the child or the parent survey.

### 2.3. Measures

#### 2.3.1. Individual Factors

Children self-reported their gender (girl vs. boy), age (years), and grade in school (4th, 5th, 6th). Parents who completed the survey provided their gender (woman vs. man), age (under 30, 30–44, 45+ years), and working status (working vs. not working). Parents reported car ownership (“Does your household have regular use of a car (including car share)?”), home ownership (“Does your family own your home?”), and highest education level (high school or less vs. college/university). Parents also indicated the number of children (≤10 years) and teens (11–15 years) within the household (sibling vs. no sibling), language spoken at home (“Do you speak a language other than English or French at home?”, yes vs. no), method of travel to work (“How do you usually travel to and from work?”, bike/public transit/car/walk, and more than one option could be selected), whether their child had “a long-standing illness, disability, or infirmity” (illness vs. no illness), and whether their child had a mobile phone (yes vs. no).

#### 2.3.2. Social Environment Factors

Social environment questions were drawn from Shaw and colleagues’ 16-country CIM study [20]. Parents were asked about their perceptions of informal social control, stranger danger, and traffic danger. Parents were asked to what extent they agreed or disagreed with the statements “Most adults who live in the neighborhood look out for other people’s children in the area” and “Some young people and adults in the area make you afraid to let your children play outdoors” on a 5-point Likert scale (1 = strongly disagree and 5 = strongly agree). Parents were also asked, “How worried are you about the risk of your child being injured in a traffic accident when crossing a road?” and responded on a scale from 1 to 4 (1 = not at all, 4 = very).

Parental perceptions of barriers to children walking and cycling were assessed using seven items that applied to the school route and more broadly to the neighborhood. Parents indicated their level of agreement with the statements “There are no sidewalks or bike lanes”, “The route does not have good lighting”, “There is too much traffic around our home”, “There is one or more dangerous crossing”, “It is unsafe because of crime (strangers, gangs, drugs)”, and “My child gets bullied, teased, harassed” on a scale of 1 to 4 (1 = strongly disagree, 4 = strongly agree).

Children were asked about their perceptions of neighborhood safety, “How safe do you feel on your own in your local neighborhood?”, on a 4-point scale (1 = not at all safe, 4 = very safe). Additionally, children were asked about their concerns (“When you are outside on your own or with friends are you worried by any of the following?”) in relation to each of the following items: “Traffic”, “Getting lost”, “Bullying”, “Strangers”, “Do not feel that I am old enough to go about on my own”, and “Not knowing what to do if someone speaks to me” (yes vs. no).

#### 2.3.3. Physical or Built Environment Factors

The built environment was assessed by site (Ottawa, Vancouver, Trois-Rivières) and objective measures of urbanization (urban, suburban, rural) as categorical variables following the methods outlined in Rainham et al. [29], school-level SES (high vs. low) estimated from 2006 Canada census data on median household income within the census tract of the school, and the neighborhood walkability of the environment within network buffers of 400 m and 1600 m. The neighborhood walkability was assessed for the child’s home address with a modified version of the index [30] using ArcGIS 10.3 (ESRI Canada, Ottawa, ON, Canada). Canadian geographic information systems do not readily contain retail floor area ratio information compared to other countries [31]. The formula used for walkability was walkability = [(2 × z-intersection density) + (z-net residential density) + (z-land use mix)].

#### 2.3.4. Independent Mobility Measures

CIM was assessed through parent self-report. CIM was operationalized as Hillman’s six mobility licences which included children’s licence to (1) travel home from school alone, (2) cross main roads alone, (3) cycle on main roads alone, (4) travel on buses alone (other than school bus), (5) travel alone to places other than school, and (6) go out alone after dark [11]. The six CIM licences were dichotomized (1 = yes; 0 = no), and a CIM index was constructed ranging from 0 to 6 as the sum of scores for each mobility licence (0 = no independent mobility and 6 = high independent mobility). A separate ATIM pilot study by Larouche and colleagues [32] found the CIM index to be reliable and valid for both English and French surveys.

### 2.4. Statistical Analysis

The “mice” package in R Studio software [33] was used to impute the incomplete multivariate data [34]. A total of 20 imputed datasets was produced with 25 iterations per imputation. Predictive mean matching was used for continuous data, logistic regression was used for binary data, proportional odds were used for ordered categorical data, and polytomous logistic regression was used for unordered categorical data [34]. Statistical analyses were performed using IBM SPSS Statistics version 25 [35]. The study sample was described using descriptive statistics. Linear mixed-effects models were used to examine the association of individual, social, and physical or built environmental correlates and parent-reported CIM. Maximum likelihood null models were created including site, urbanization, SES, and schools to determine the within-school intra-class correlation coefficient (ICC), resulting in school being a significant random effect. Site, urbanization, and SES were therefore assigned as fixed effects in all models. The models corresponded to the levels of a social–ecological model: Model 1 examined the individual correlates and CIM, Model 2 examined the social environment correlates and CIM, and Model 3 examined the geographical and area-level correlates of CIM. Child age and child grade violated the assumption of multicollinearity; therefore, child grade (*β* = 0.513) was included as a proxy for child age. Subsequent analyses were stratified by gender due to well-documented gender differences in CIM [16,36,37]. Both non-stratified and stratified results are provided in Section 3. A significance level of *p* < 0.05 was used for all statistical inferences. Finally, the multiply imputed models were compared with a complete case analysis to examine the consistency of the findings.

## 3. Results

Table 1 presents the descriptive characteristics of 1699 participants. Over half (55.1%; *n* = 936) of child participants were girls, and children’s ages ranged between 8 and 13 years (mean age = 10.21 ± 0.98 years). More than three-quarters (80.9%; *n* = 1375) of parent respondents were women, in the age range of 30–44 years (68.9%), working either full-time or part-time (84.5%), and held a college or university degree (87.8%). Using the parent-reported CIM index (0–6 scale), the mean independent mobility index was 2.06 ± 1.55. Boys’ independent mobility (mean = 2.16 ± 1.56) was significantly higher than girls’ independent mobility (mean = 1.98 ± 1.54, *t =* 2.341, *p =* 0.019). On the independent mobility index, 21.0% of children ranked 0, 20.4% ranked 1, 18.1% ranked 2, 18.4% ranked 3, 17.0% ranked 4, 4.8% ranked 5, and only 0.4% ranked 6.

Table 2 shows the individual-level correlates of CIM. Prior to stratification by gender, children’s grade in school (*β* = 0.612, *p* < 0.001), their gender (ref. boy, *β* = −0.257, *p* < 0.001), and phone ownership (*β* = 0.593, *p* < 0.001) were all significantly associated with CIM. Additionally, language spoken at home other than English or French (*β* = −0.503, *p* < 0.001), parent gender (ref: man, *β* = −0.269, *p* = 0.001), and car ownership (ref: no car, *β* = −0.374, *p* < 0.05) were negatively associated with CIM. For both girls and boys, grade in school (*β* = 0.560, *p* < 0.001 for boys; *β* = 0.658, *p* < 0.001 for girls), language spoken at home (*β* = −0.599, *p* < 0.001 for boys; *β* = −0.487, *p* < 0.001 for girls), and ownership of a mobile phone (*β* = 0.433, *p* < 0.01 for boys; *β* = 0.700, *p* < 0.001 for girls) remained significant after stratifying by child gender. Boys had significantly higher independent mobility if the parent respondent traveled to work by car (*β* = 0.271, *p* < 0.05). However, the gender of the parent respondent (i.e., mother respondent) was negatively associated with boys’ independent mobility (*β* = −0.387, *p* < 0.05).

Table 3 shows social-environment-level correlates of CIM. Prior to stratification by gender, a child’s worry about getting lost (*β* = −0.2.75, *p* < 0.05) was negatively associated with CIM. Parents’ worry regarding the risk of their child being injured in a traffic accident was negatively associated with CIM (*β* = −0.321, *p* < 0.001). Additionally, parents’ concerns that their neighborhood was unsafe due to crime (strangers, gangs, drugs) (*β* = −0.235, *p* < 0.001), that there were one or more dangerous crossings present (*β* = −0.209, *p* < 0.001), or that there were no sidewalks or bike lanes (*β* = −0.114, *p* < 0.05) were negatively associated with CIM. Parental perceptions that their child might face bullying, teasing, or harassment were positively associated with CIM (*β* = 0.190, *p* < 0.05). Stratifying by gender showed that parents’ concern over traffic (*β* = −0.287, *p* < 0.001 for boys; *β* = −0.339, *p* < 0.001 for girls) and the presence of one or more dangerous crossings (*β* = −0.176, *p* < 0.01 for boys; *β* = −0.292, *p* < 0.001 for girls) was negatively associated with independent mobility in boys and girls. For boys, parents’ perception of crime (strangers, gangs, drugs) in the neighborhood was negatively associated with CIM (*β* = −0.313, *p* < 0.001). For girls, parents’ concern over stranger danger (*β* = −0.146, *p* < 0.05) and a child’s worry about getting lost (*β* = −0.364, *p* < 0.05) were negatively associated with CIM. Parental concern regarding the absence of sidewalks or bike lanes and bullying was no longer significant after stratification.

Table 4 shows geographical and area-level correlates of CIM. Site and urbanization were significantly associated with CIM. Compared to children from Trois-Rivières, children in Ottawa (*β* = −1.188, *p* < 0.001) and Vancouver (*β* = −1.216, *p* < 0.001) had lower CIM. In comparison to rural environments, suburban environments were negatively associated with CIM (*β* = −0.382, *p =* < 0.05). No significant associations were found for children living in urban versus those in rural areas. These results remained consistent for site after looking at boys and girls separately (Ottawa: *β* = −0.979, *p* < 0.001 for boys; *β* = −1.273, *p* < 0.001 for girls; Vancouver: *β* = −1.058, *p* < 0.001 for boys; *β* = −1.295, *p* < 0.001 for girls). After stratifying by gender, suburban environments, in reference to rural areas, were negatively associated with independent mobility for boys (*β* = −0.536, *p* < 0.05) but not for girls (*β* = −0.217, *p* = 0.275). For girls, neighborhood walkability (400 m buffer) was positively associated with CIM (*β* = 0.064, *p* < 0.01) but negatively associated with CIM with a 1600 m buffer (*β* = −0.059, *p* < 0.05). Additionally, area-level SES was not significantly associated with CIM (*β* = −0.129, *p =* 0.385).

The examination of multiply imputed models compared to complete case analysis found comparable results, with a few notable differences primarily concerning individual correlates of CIM. A child having a long-standing illness, disability, or infirmity was negatively associated with CIM (*β* = 0.388, *p* < 0.05). Parental travel mode to work was significant for boys’ and girls’ CIM. For boys, parents traveling to work via public transport was negatively associated with CIM (*β* = −0.518, *p* < 0.05). For girls, parents traveling to work via cycling was positively associated with CIM (*β* = 0.478, *p* < 0.05). Additionally, parental work status (either part- or full-time) was positively associated (*β* = 0.530, *p* < 0.01) and parent age was negatively associated with girls’ independent mobility (*β* = −0.328, *p* < 0.01). Additionally, a child’s perception of neighborhood safety was positively associated with CIM (*β* = 0.209, *p* < 0.001). The multiply imputed data results are displayed in Table 2, Table 3 and Table 4, and complete case analyses results (Appendix A) are in the Appendix A.

## 4. Discussion

The aim of this study was to examine factors influencing CIM amongst three distinctly different sites across Canada each with varying urbanization (urban, suburban, rural) and SES (high vs. low) environments. Unsurprisingly, children’s individual characteristics, specifically grade level in school and gender, were significantly associated with CIM, in line with previous literature indicating that older children were more likely to have higher levels of CIM [38,39,40]. As a child gains maturity, knowledge, and pertinent skills, parents feel more comfortable letting the child roam independently. Also, children’s gender predicted CIM such that boys were more likely to have higher levels of CIM, consistent with previous literature [25,40,41,42,43]. Future research may wish to examine more closely how gender within the family unit (e.g., mothers, fathers, sons, daughters) may influence perceptions of the social environment and, consequently, CIM. Car ownership was negatively associated with CIM, indicating that increased car ownership or access to a vehicle negatively impacts children’s levels of independent mobility and active travel [24,43,44].

Language spoken at home (when different from the co-official Canadian languages English and French) was significantly negatively associated with CIM. Language spoken at home may reflect social and cultural norms, which may affect CIM by influencing parental decision-making [45]. Studies in New Zealand [46] and the United States [47] found differences in CIM based on race and/or ethnicity. These differences may influence household make-up (e.g., single-parent household), community make-up (e.g., extended family living in close proximity) [46], parental concerns about the neighborhood environment [48], and commuting mode [49], which may in turn impact CIM. As Wolfe and colleagues [47] argue, interventions should consider the “social and cultural norms of different races and ethnicities” to better predict how active travel plans will be received by a diverse array of families (p. 977). However, other studies have found no significant association between ethnicity and/or race and CIM [49,50]. Regardless of mixed findings, it may be important to further examine the influence of race and/or ethnicity and, by extension, social and cultural norms that may impact CIM. Interestingly, mobile phone ownership was a significant factor associated with both girls’ and boys’ independent mobility. A child’s ownership of a mobile phone may give parents a sense of security and social control [51], a way to communicate amongst family members [52], and a tool for long-range surveillance of children.

Several social environmental factors were significantly associated with both girls’ and boys’ independent mobility. Parents’ perceptions and concerns regarding traffic danger, crime, and dangerous crossings were, unsurprisingly, negatively associated with CIM. These findings echo previous literature showing lower levels of independent mobility when parents are concerned about the neighborhood environment and perceived danger from traffic, crime, and the built environment (e.g., dangerous crossings) [40,41,50,53]. These results suggest that regardless of gender, real and perceived dangers from traffic, crime, and the built environment negatively influence CIM. Additionally, a child’s own worry about getting lost was a factor negatively associated with independent mobility. For girls, a child’s worry about getting lost was still significant after stratifying by gender.

For boys, independent mobility levels were higher when the parent respondent traveled to work by car but lower when parents perceived crime within the neighborhood and when the parent respondent was a mother. Past research has found a positive association between parents’, especially mothers’, increased working hours and longer distances to work and their child’s independent mobility [49]. Car usage, longer distances to work, or work destination in the opposite direction to a child’s school may prompt children to travel independently (e.g., public transit) or actively (e.g., walking, cycling) to school or other destinations. Also, prior to stratification by gender, parents’ perceptions that a child might face bullying, teasing, or harassment were positively associated with boys’ independent mobility, which may not intuitively make sense. This may reflect parental perceptions that such a possibility is heightened, rather than real, given their child’s increased exposure to other children while unsupervised by an adult.

A notable contribution of this study given the sampling frame is that several physical or built environmental correlates were not significantly associated with CIM. Area-level SES was not related to CIM in this study. Findings regarding the association between CIM and SES have been mixed in past research. While some studies have found no association with SES [24,47,54], others have found low- and middle-SES environments to be more conducive for CIM [38,49]. Additionally, the level of urbanization was not significant, except for boys living in suburban environments showing a lower level of CIM compared to boys living in rural environments. Previous literature on urbanization and CIM have found mixed results regarding the suburban environment having a positive association [44,55] with CIM compared to urban and rural environments. In Kytta’s model explaining the covariation of independent mobility and actualization of affordances in four different environments (i.e., Bullerby, Wasteland, Cell, and Glasshouse), some suburban environments can be “sleepy” or “too dull” and may be categorized as “Wasteland” environments [56]. Affordances are defined as opportunities (e.g., physical, emotional, social, and cultural) which an individual perceives within a specific environment, while actualized affordances are ones that “the individual perceives, utilizes or shapes” [56], p. 181. Suburban environments may provide a lack of diversity of affordances and may be empty of things for children to discover, thereby limiting actualized affordances [56].

Neighborhood walkability was only associated with girls’ independent mobility. These findings are similar to two studies which found walkability to be positively associated with CIM, but only in girls [53,57]. However, walkability has yielded mixed results, including no significant association between parental perceptions of walkability and CIM [41]. Walkable neighborhoods for adults may not be supportive of physical activity among children [58]. Higher intersection density, for example, will result in a greater number of street crossings. CIM may be restricted as a result of parental safety concerns about the subsequent increase in potential exposure of their child to traffic. However, in the current study, walkability within the 400 m buffer was positively associated with girls’ CIM, but not when using a larger buffer (1600 m). It is well documented in the literature that increasing distances, especially above 1 km, result in lower levels of CIM [23,48,49]. Accordingly, the walkability of the area most proximal to the home appeared important for girls.

Overall, the strongest correlate of CIM, even after stratification by child gender, was location (i.e., site). CIM was lower in Ottawa and Vancouver compared to Trois-Rivières. While populations in these locations (Ottawa = 934,243; Vancouver = 631,486; Trois-Rivières = 134,413 according to 2016 Census Profile) [59] may influence CIM (e.g., population density), other factors like the social and cultural differences should be considered. Secondary analyses (not reported here) found no significant associations between site and parents’ perceptions of informal social control, traffic concerns, and stranger danger. The differences in CIM by site may stem from social and cultural differences.

Additionally, as a multi-cultural country, Canada is home to a diverse array of people, especially in hubs likes Ottawa and Vancouver, and therefore encompasses a range of cultural and social norms. This is reflected by the diversity of languages spoken at the two larger sites, Vancouver and Ottawa, compared to Trois-Rivières. Secondary analyses determined that higher proportions of languages (other than English and French) were spoken at home in Ottawa and Vancouver compared to Trois-Rivières. In Trois-Rivières, only 4.1% reported speaking a language other than English or French, while this percentage was 32.5% and 47.0% for Ottawa and Vancouver, respectively. In Ottawa, higher percentages of Arabic, Creole, and Spanish were spoken at home, while in Vancouver, higher percentages of Asian languages were spoken (i.e., Mandarin, Chinese, Cantonese, Korean, and Japanese). Participants in Ottawa and Vancouver self-reported speaking over 45 different languages at home compared to 7 self-reported languages in Trois-Rivières.

Language differences may reflect cultural differences. Potential differences in cultural norms may influence family structure and social norms regarding CIM, which may in turn affect whether families adopt or reject certain travel modes (e.g., independent travel). In a study by Lam and Loo in Hong Kong [60], higher numbers of grandparents or domestic helpers within the family structure reduced children’s independent travel opportunities, and children from extended households showed lower levels of CIM compared to children from nuclear families or single-parent families. These cultural differences can be seen globally for CIM as well as active transportation [61]. Although CIM has declined worldwide, Scandinavian countries like Finland, Norway, Sweden, and Denmark, as well as other countries like Germany and Japan, rank the highest for CIM [20]. The most recent Global Matrix 3.0, the most comprehensive assessment of global variation in children and youth physical activity, found that while Canada and the United States scored a D− for active transportation, other countries such as Japan, Nepal, Denmark, and Finland scored in the A− to B+ range [62]. It is therefore necessary to consider how social and cultural differences may influence CIM and consequently develop research, strategies, and policies that are tailored and take these differences into account.

While the physical or built environment and individual-level variables are strong correlates of CIM, many of these factors can also be classified as *non-modifiable* factors, such as location and child age. In terms of location, levels of CIM looked remarkably similar regardless of where a child lived. In the hopes of positively influencing CIM levels, it may be important to shift focus to *modifiable* factors. These modifiable factors encompass the social correlates of CIM, more specifically, parents’ and children’s perceptions of safety. After stratification by gender, the majority of the significant correlates were parental perceptions of the social environment. In line with previous literature, this study found that parental perceptions of neighborhood safety, crime, bullying, stranger danger, and traffic were significantly associated with CIM [25,36,41,50].

These findings may have implications for policies and interventions that aim to encourage CIM, active transportation, and outdoor play. While city planners and urban developers can work toward creating child-friendly environments, it is important to acknowledge parents’ role as “gatekeepers” for their children’s access to the outside world. Parental influence over their CIM licenses may be considered a social environmental influence and may play a role in either enhancing or restricting children’s actual mobility. Interventions that aim to increase CIM will need to target parents’ fears and concerns (e.g., perceptions of traffic danger, neighborhood safety) as these are modifiable factors that can be addressed. There are current initiatives that aim to help parents reframe those risks. For example an online tool, OUTSIDEPLAY.ca, developed by researchers at British Columbia Children’s Hospital, The University of British Columbia, BC Injury Research and Prevention Unit, aims to address parental fears by helping them reframe the risk and gain confidence in allowing their children to engage in outdoor risky play [63]. The social–ecological framework emphasizes the interplay of several layers of influence on a health behavior (e.g., CIM); while we suggest focusing research on modifiable factors, it is still important to consider how to improve neighborhood safety.

Finally, in line with previous literature, significant differences in CIM are seen between boys and girls. Additionally, this study found a negative association between parent gender (i.e., mother respondents) and boys’ independent mobility. Previous research has noted differences between fathers and mothers regarding risk allowance and negotiation for recreational activities. Fathers are often deemed “risk experts” while mothers tend to be more protective and tend to counter the father’s risk allowance [64], p. 1390. The findings identify a need for future research to more closely explore the gendered nature of CIM.

### Strengths and Limitations

The strengths of this study include the relative large sample size (*n* = 1699), as well as sample stratification by region (ON, BC, and QC), urbanization (urban, suburban, and rural), and socioeconomic status (high vs. low median income), although more than half the parent respondents were women (80.9%) and most were highly educated (87.8%). Additionally, while attempts were made to vary the regions and urbanization where the sample was recruited, it is important to acknowledge that not all environments were considered. For example, people living in northern regions of Canada may face unique barriers to CIM and physical activity, such as wildlife, inclement weather, and hours of daylight available. The questions examining social environment factors were drawn from a 16-country study; however, it is important to acknowledge that there are no published data on the reliability and validity of these measures. This study used the most common and validated measure of CIM [65], yet the responses may be vulnerable to recall and social desirability biases.

## 5. Conclusions

CIM is influenced by a diverse set of correlates including individual, social, and physical or built environmental level factors. While there are non-modifiable factors including individual and physical or built environmental factors that influence CIM, it will be vital for interventions to target modifiable factors, including children’s and parents’ perceptions of their social environment. Perceptions of neighborhood safety (e.g., traffic, crime, and stranger danger) can be influenced and may offer a target area for CIM interventions. Moreover, the influence of gender and cultural background needs to be further examined in order to help address parents’ perceptions of safety, concerns, and worries, which in turn can affect CIM.

## Figures and Tables

**Table 1 ijerph-16-02862-t001:** Descriptive characteristics of the study sample (*n* = 1699).

Parent Characteristic	*N*	Percentage (%)	Child Characteristic	*N*	Percentage (%)
**Gender**			**Gender**		
Woman	1375	80.9	Girl	936	55.1
Man	324	19.1	Boy	763	44.9
**Age (years)**			**Age (years)**		
Under 30	19	1.1	8	15	0.9
30–44	1170	68.9	9	440	25.9
45+	510	30.0	10	584	34.4
			11	493	29.0
			12	161	9.5
			13	6	0.4
**Education Level**			**Grade Level**		
High school or less	166	9.8	4	582	34.3
College/University	1491	87.8	5	600	35.3
**Language Spoken at home**			6	498	29.3
Yes, speak a language other than English or French at home	517	30.4	7	19	1.1
**Parent Work Status**			**Child Illness**		
No	263	15.5	Yes, child has a long-standing illness, disability, or infirmity	80	4.7
Yes, full-time or part-time	1436	84.5	**Child mobile phone ownership**		
**Car Ownership**			Yes	227	13.4
No car	62	3.6			
Yes, own 1 or more cars	1637	96.4			
**Home Ownership**					
No, do not own home	447	26.3			
Yes, own home	1252	73.7			
**Siblings**					
No	285	16.8			
Yes	1412	83.1			
**Parent Travel Mode to Work**					
Walk	221	13.0			
Bike	108	6.4			
Public Transit	208	12.2			
Car	1200	70.6			

**Table 2 ijerph-16-02862-t002:** Individual-level correlates of children’s independent mobility.

Correlate	Girls (*n* = 936)		Boys (*n* = 763)			
	*β*	95% CI		*β*	95% CI	
**Child Characteristics**						
Child grade level	**0.658** ***	0.550	0.766	**0.560** ***	0.433	0.688
Child illness	−0.278	−0.716	0.159	−0.255	−0.686	0.176
Mobile phone ownership	**0.700** ***	0.452	0.948	**0.433** **	0.117	0.750
**Household Characteristics**						
Parent age	−0.169	−0.360	0.022	0.015	−0.200	0.229
Parent gender	−0.160	−0.375	0.055	**−0.387** **	−0.634	−0.140
Parent work status (not working vs. working)	0.177	−0.083	0.438	−0.102	−0.397	0.193
Parent education	0.011	−0.268	0.289	−0.160	−0.510	0.190
Language spoken (English/French vs. other language)	**−0.487** ***	−0.712	−0.262	**−0.599** ***	−0.845	−0.354
Car ownership	−0.445	−0.910	0.019	−0.179	−0.750	0.393
Home ownership	−0.038	−0.252	0.176	0.204	−0.054	0.463
Siblings (no sibling vs. sibling(s))	0.043	−0.193	0.279	0.075	−0.182	0.333
**Parent Travel Mode to Work**						
Walk	−0.123	−0.394	0.148	0.309	−0.017	0.636
Bike	0.351	−0.004	0.706	0.045	−0.375	0.465
Public transit	−0.228	−0.504	0.047	0.060	−0.255	0.376
Car	−0.059	−0.266	0.148	**0.271** *	0.022	0.519

Significant correlates are bolded: * *p* < 0.05, ** *p* < 0.01, *** *p* < 0.001; CI: confidence interval; *β*: unstandardized regression coefficients.

**Table 3 ijerph-16-02862-t003:** Social-environment-level correlates of children’s independent mobility.

Correlate	Girls (*n* = 936)			Boys (*n* = 763)		
	*β*	95% CI		*β*	95% CI	
**Child Perceptions**						
Neighborhood safety	−0.194	−0.405	0.017	0.025	−0.186	0.235
**Child worried about…**						
Traffic	−0.102	−0.419	0.215	−0.175	−0.497	0.147
Getting lost	**−0.364** *	−0.670	−0.059	−0.279	−0.602	0.044
Bullying	0.248	−0.115	0.612	0.175	−0.176	0.527
Strangers	−0.194	−0.471	0.083	0.001	−0.272	0.273
Feeling they are not old enough to go about on their own	−0.244	−0.646	0.159	−0.231	−0.642	0.181
Not knowing what to do if someone speaks to them	0.060	−0.235	0.356	−0.264	−0.572	0.044
**Parent Perceptions**						
Most adults in the neighborhood look out for other people’s children in the area	0.070	−0.051	0.192	0.059	−0.065	0.183
People in the area make me afraid to let my child play outdoors	**−0.146** *	−0.268	−0.023	−0.034	−0.156	0.088
Worried about risk of child being injured in a traffic accident	**−0.339** ***	−0.494	−0.183	**−0.287** ***	−0.441	−0.134
**Barriers to child walking or cycling**						
No sidewalks or bike lanes	−0.093	−0.242	0.056	−0.117	−0.261	0.026
Route does not have good lighting	0.139	−0.029	0.307	0.069	−0.089	0.228
Too much traffic around the home	−0.007	−0.152	0.139	0.027	−0.120	0.174
One or more dangerous crossing	**−0.292** ***	−0.423	−0.162	**−0.176** **	−0.307	−0.046
Unsafe due to crime (strangers, gangs, drugs)	−0.144	−0.302	0.014	**−0.313** ***	−0.473	−0.152
Child gets bullied, teased, harassed	0.263	−0.001	0.527	0.158	−0.079	0.396

Significant correlates are bolded: * *p* < 0.05, ** *p* < 0.01, *** *p* < 0.001; CI: confidence interval; *β*: unstandardized regression coefficients.

**Table 4 ijerph-16-02862-t004:** Geographical and area-level correlates of children’s independent mobility.

Correlate	Girls (*n* = 936)			Boys (*n* = 763)		
	*β*	95% CI		*β*	95% CI	
Socioeconomic status	−0.182	−0.486	0.122	−0.106	−0.472	0.260
**Site**						
Ottawa, ON, Canada	**−1.273** ***	−1.657	−0.890	**−0.979** ***	−1.443	−0.514
Vancouver, BC, Canada	**−1.295** ***	−1.660	−0.929	**−1.058** ***	−1.495	−0.622
Trois-Rivières, QC, Canada	0	.	.	0	.	.
**Urbanization**						
Urban	−0.154	−0.589	0.282	−0.318	−0.816	0.180
Suburban	−0.217	−0.612	0.178	**−0.536** *	−1.004	−0.068
Rural	0	.	.	0	.	.
**Walkability**						
400 m	**0.064** *	0.015	0.114	−0.003	−0.062	0.055
1600 m	**−0.059** *	−0.114	−0.003	0.024	−0.035	0.083

Significant correlates are bolded: * *p* < 0.05, ** *p* < 0.01, *** *p* < 0.001; CI: confidence interval; *β*: unstandardized regression coefficients; m: meters.

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
