# Peer review of "Correlates of Children’s Independent Mobility in Canada: A Multi-Site Study"

_ijerph, 2019, doi:10.3390/ijerph16162862_

Round 1

Reviewer 1 Report

This is an interesting, topical and clearly written study. Declining PA and CIM is an important contemporary public health challenge. Overall I think the paper is well written, my main concern is regarding the novelty of this study and what the findings add to the literature - I believe the discussion can be strengthened to convey the novelty of the findings. 

Abstract: Children’s independent mobility, is this a behaviour or opportunity for intervention? I don’t believe this is a context..consider rewording. Please include the age of included children in the abstract. Re conclusion – could environmental factors e.g neighbourhood walkability not also be targeted for future intervention?

Introduction:

Page 2 line 46 – The authors mentioned physical activity levels have failed to increase despite efforts to encourage higher rates of PA– Could you provide some context here regarding the types of interventions or approaches that have been used to date that have been largely unsuccessful? This may provide some further justification for the variables you have assessed. 

Line 49 – Why was CIM selected as a target area? Can you provide more information for this, at this point you have not identified the age group you are referring to which is important to consider CIM, as children need to be at an age where they have some agency to engage in CIM.  

Methods – Methods are well described. Missing details of sample size calculation, did you obtain enough participants? Can you include and ethical approval number? Was information collected regarding parental relationship status or employment status of the non-responding carer (if applicable). These factors could mediate the relationship between CIM and parental gender.

The relationship between location, child gender, age and parent gender/ethnicity and CIM have been explored in previous research. More detail is needed in your discussion of ethnicity to ensure that your study is adding to existing literature. 

As well as addressing modifiable social factors e.g. perception of risk – could these be used in combination with interventions for the physical environment – to also improve neighbourhood safety. There are a multitude of factors that shape a parental concern for their child’s safety in relation to physical activity including previous experiences with injury, contextual factor including incidents in the community, parent’s own upbringing, the number of children they have. Given the diversity of influencing factors, targeting parent’s safety concerns in isolation without altering physical neighbourhood safety and a sense of community 

Author Response

Dear Reviewer 1,

Thank you for your feedback and suggestions. Please see the attachement.

Thank you.

Reviewer 2 Report

Correlates of children’s independent mobility in Canada: A multi-site study

Thank you for the opportunity to review this manuscript. The current article estimates individual, social and built environmental correlates of children’s independent mobility in a study population of three regions in Canada. The study addresses an interesting and worthwhile topic, it is well conceptualized and provides an in-depth view on the growing research topic of children’s independent mobility assessed through parental license.

Below I made some suggestions to improve the manuscript:

Minor Comments

Whole Paper

The term “environmental correlates” is used in the whole manuscript in order to describe built and physical environmental correlates. In line with the social-ecological framework, I suggest to use the term “physical or built environmental correlates” instead of “environmental correlates”, because it could also include the social environment (like social norms or social support from others).

Abstract

The abstract needs a revision due to punctuation and grammar (e.g. line 27). Additionally, the first sentence could be improved by a statement for introducing the study issue and the aim of the study which also should be mentioned in the abstract. 

In lines 30 and 31 “boys’ CIM” and “girls’ CIM” need rewording.

Introduction

Line 43 and line 46: citation no. [1] is probably incorrect and should be perhaps no. [8]

Line 52: citation no. [9] does not contain any definition of children’s independent mobility

Lines 60/61: It is not clear what “significant restrictions” are referred to.

Methods

In the methods section the recruitment of participants in grade 4, 5, and 6 is described, however, in the result section grade levels up to 7 are mentioned. The same is reported for the age of the participants: In the introduction and method sections the participants are described as 10-12 year-old children. However, in the result section an age range of 8 to 13 years is reported (Table 1). Please clarify the different 

Lines 119-138: The provision of information on reliability and validity of the measures of the social environment factors could improve the manuscript?

Discussion

Lines 287-289: A positive significant association of boys’ independent mobility and parents’ perception that a child might face bullying, teasing, or harassment is not reported in the result section (Table 3), however, it is discussed. Please clarify!

Independent mobility was assessed via a well-known and typically used questionnaire assessing parental license for independent mobility. However, this measure is not meant to reflect children’s actual mobility as a health-enhancing behavior. In line with the social-ecological framework it can also be seen as a social environmental factor (parental influence). Parental license could be a determinant that restricts or enhances children’s actual mobility. This aspect should be added in the discussion.

Conclusion

In the conclusion section, the authors state that environmental factors are non-modifiable (line 392). First, same as mentioned above, the term “environmental factors” include the social environment. Secondly, this statement is too general as there are physical environmental factors, which are modifiable. Please be precise and state which physical or built environmental factors are non-modifiable.

Author Response

Dear Reviewer 2,

Thank you for your feedback and suggestions. Please see the attachement.

Thank you.
